# Fertility-Sparing Treatment for Endometrial Cancer: Oncological and Obstetric Outcomes in Combined Therapies with Levonorgestrel Intrauterine Device

**DOI:** 10.3390/cancers14092170

**Published:** 2022-04-26

**Authors:** Ida Pino, Anna Daniela Iacobone, Ailyn Mariela Vidal Urbinati, Maria Di Giminiani, Davide Radice, Maria Elena Guerrieri, Eleonora Petra Preti, Silvia Martella, Dorella Franchi

**Affiliations:** 1Preventive Gynecology Unit, European Institute of Oncology IRCCS, 20141 Milan, Italy; annadaniela.iacobone@ieo.it (A.D.I.); ailyn.vidalurbinati@ieo.it (A.M.V.U.); mariaelena.guerrieri@ieo.it (M.E.G.); eleonora.preti@ieo.it (E.P.P.); silvia.martella@ieo.it (S.M.); dorella.franchi@ieo.it (D.F.); 2Department of Biomedical Sciences, University of Sassari, 07100 Sassari, Italy; 3Unit of Obstetrics and Gynecology, ASST Fatebenefratelli-Sacco, Department of Biological and Clinical Sciences L. Sacco, University of Milan, 20157 Milan, Italy; maria.digiminiani@unimi.it; 4Division of Epidemiology and Biostatistics, IEO European Institute of Oncology IRCCS, 20141 Milan, Italy; davide.radice@ieo.it

**Keywords:** endometrial cancer, fertility-sparing treatment, obstetric outcomes, oncological outcomes

## Abstract

**Simple Summary:**

This article discusses a retrospective study describing sixteen years of experience in the fertility-sparing treatment (FST) of endometrial cancer (EC) in a tertiary referral center for oncology. The aim of the study is to compare oncological and reproductive outcomes of different combined therapy with LNG-IUD in FST of presumed FIGO STAGE IA endometrioid G1 EC. We assessed outcomes for 75 patients treated with three different approaches: GnRH analogue (GnRHa) + LNG-IUD vs. Megestrol acetate (MA) + LNG-IUD vs. MA + LNG-IUD + Metformin (MET). We reported, although not statistically significant, an increasing rate of CR from the regimen with GnRHa to one with MA+MET (65% vs. 83%) and showed a statistically significant lower risk of recurrence in women treated with MA+ LNG-IUD+MET when compared to GnRHa+ LNG-IUD regimen. There were no differences in obstetric outcomes among different therapeutic regimens.

**Abstract:**

Background: The prevalence of reaches up to 5% in women younger than 40 years. Therefore, the fertility preservation should be the goal of the clinical practice in women with desire of pregnancy and low-risk features. The aim of this study is to compare oncological and reproductive outcomes of different hormonal therapies in FST of EC. Methods: A retrospective single-center study recruiting patients with presumed FIGO STAGE IA endometrioid G1 EC from 2005 to 2020 was performed. We assessed outcomes for three different therapeutic options: GnRHa + LNG-IUD vs. MA + LNG-IUD vs. MA + LNG-IUD + MET. Results: In total, 75 patients were enrolled and followed up for a median of 45 months. Complete response (CR) was achieved in 75% of patients at 12 months. Although not statistically significant, we reported an increasing rate of CR from the regimen with GnRHa to the one with MA + MET (65% vs. 83%). We showed a statistically significant lower risk of recurrence in women treated with MA + LNG-IUD + MET, when compared to GnRHa + LNG-IUD regimen. The pregnancy rate was 74% and live birth rate was 42%, with no differences among regimens. Conclusions: FST is a safe and effective option in women who desire to preserve fertility.

## 1. Introduction

Endometrial cancer (EC) is the most common gynecologic tumor in developed countries and the fifth cause of cancer in women worldwide [1,2]. Although typically considered as a postmenopausal cancer, between 15% and 25% of cases are detected in premenopausal women, 3–5% of whom are younger than 40 years [3].

The occurrence of EC at young age has been associated with prolonged estrogen exposure not opposed by progesterone, that can result from obesity, polycystic ovary syndrome (PCOS), infertility, and nulliparity [4]. Several studies also suggested that insulin resistance and hyperinsulinemia are associated with a higher risk of EC. Insulin has a mitogenic and antiapoptotic activity and shares downstream signaling pathways with insulin-like growth factor-1, contributing to endometrial proliferation [5]. Lynch syndrome accounts for 3% of EC with a mean age of occurrence lower than general population and a lifetime risk of EC from 40% to 60% [6].

The standard treatment of EC includes total hysterectomy, bilateral salpingo-oophorectomy, and pelvic washing; with or without lymph node staging. However, fertility-sparing treatment (FST) represents an alternative strategy in selected young patients who desire to preserve fertility.

According to the recent European Society of Gynaecological Oncology (ESGO)/European Society for Radiology and Oncology (ESTRO)/European Society for Pathology (ESP) consensus, the following criteria should be considered to initiate a fertility-sparing management of EC: grade 1 endometrioid endometrial carcinoma; disease limited to the endometrium on Magnetic Resonance Imaging (MRI) or transvaginal ultrasound (TVUS); absence of metastatic disease on imaging; no genetic risk factors [7].

Standard regimens for FST are based on progestogens, given both orally (Medroxyprogesterone acetate (MPA) 500–600 mg daily or MA 160–320 mg daily) and/or via an LNG-IUD, which directly counteracts the proliferative action of estrogens on endometrium. Alternative strategies include administration of GnRHa and/or aromatase inhibitors. Furthermore, interest is growing around new therapeutic strategies. Indeed, the use of MET, a biguanide that acts as an insulin-sensitizer agent, is currently under investigation. MET has been shown to have various anti-proliferative effects in endometrial cancer, including reduction of IGF-1 levels, increase in progesterone receptor expression, and regulation of different pathways involved in cell proliferation [8,9]. According to the current literature, adding MET to the standard progestin treatments increases both response rates and relapse-free survival [10], but seems to be more effective in overweight women and in patients with atypical endometrial hyperplasia [11,12]. The efficacy of MET has not been confirmed from all studies; randomized controlled trials are lacking and investigations on this therapeutic agent are still underway [13].

Overall, good complete regression rates and excellent overall survival of young women affected by EC and undergoing FST have been shown in previous literature. Despite several studies and metanalysis, universal results regarding the best type and duration of treatment to achieve complete remission and the effect of different regimens on pregnancy rate are still lacking.

The aim of this study is to compare oncological and reproductive outcomes of different hormonal therapy regimens in combination with LNG-IUD in young women with EC undergoing FST.

## 2. Materials and Methods

All women affected by presumed FIGO stage IA well-differentiated endometrioid EC and treated with FST at the Preventive Gynaecology Unit of the European Institute of Oncology, Milan, Italy, from 2005 to 2020, were selected for a retrospective analysis. The study was approved by our Institutional Review Board and informed consent was obtained from all the subjects. Data presented in this study are available on request from the corresponding author, but are not publicly available due to patients’ privacy restrictions. The data are safely stored in a private database of the European Institute of Oncology, Milan, Italy.

Inclusion criteria were: (1) age between 18 and 43 years; (2) desire of pregnancy; (3) histological diagnosis of well-differentiated endometrioid EC; and (4) disease confined to the endometrium without myometrial invasion (stage FIGO IA). Patients were excluded in case of: (1) suspicion of myometrial invasion on TVUS or MRI; (2) histological diagnosis of moderately or poorly differentiated EC; or (3) extrauterine disease.

All patients underwent pre-treatment evaluation including (1) hysteroscopy with endometrial biopsy for histological diagnosis, or pathological review of original slides if initial diagnosis was made at different institutions; (2) TVUS and MRI to rule out myometrial invasion, extrauterine metastasis, or the presence of synchronous ovarian cancer. In case of patients referred to our Institute with a diagnosis of EC obtained with different methods, a hysteroscopic guided biopsy was repeated before the treatment. All histological diagnoses were performed by dedicated gynecological pathologists.

Treatment protocol varied over time and included three different options of hormonal therapy:GnRHa + LNG-IUD, from February 2005 to September 2012;MA 160 mg/day + LNG-IUD, from October 2012 to September 2017;MA 160 mg/day + LNG-IUD + MET 500 mg 3 times per day (1500 mg/day), from October 2017 to October 2021.

The GnRHa used was Triptorelin Acetate in monthly depot injection of 3.75 mg and the LNG-IUD was Mirena^®^ (Bayer Health Care Pharmaceutical Inc., Wayne, NY, USA).

Response to treatment was evaluated with endometrial samples collected by office hysteroscopy, allowing direct visualization of the uterine cavity and targeted biopsies, after 6 and 12 months of therapy. The final response was classified as: complete response (CR), if the final histological examination showed normal endometrial characteristics; partial response (PR) when AEH was diagnosed in patients with initial EC; stable disease (SD), in case of persistence of the same histological diagnosis; and progression of disease (PD), when women with initial well-differentiated (G1) EC developed moderately (G2) or poorly differentiated (G3) EC.

According to our institutional protocol, patients who reached two consecutive CR could try to become pregnant. On the contrary, patients who showed SD or PD at 6 or 12 months could not pursue FST and underwent standard demolitive surgical treatment.

Data regarding the principal demographic and clinical patients’ characteristics and fertility outcomes were collected by medical reports.

The patients’ characteristics were summarized either by count and percentage, or mean and Standard Deviation (SD), for categorical and continuous variables, respectively, and tabulated according to treatment. Between treatments, comparisons for continuous variables were performed using the F-test from type III Anova fixed effects model for normally distributed data, the Kruskal–Wallis test was used for non-normally distributed data. Normality of the data was checked using the Shapiro–Wilk test. Pairwise comparisons were adjusted for multiplicity. Between treatment comparisons for categorical were conducted using Fisher’s exact test. The Cumulative Incidence for relapse at 6, 12, and 24 months by treatment and by post-treatment pregnancy status, as well as the Hazard Ratio (HR) were estimated in univariate and multivariable Cox competing risk regression models. Any events precluding the observation of relapse were considered as competing events. The CI functions were compared by using Gray’s test. All tests were two-tailed and considered significant at the 5% level except those adjusted for multiplicity. All analyses were performed using SAS 9.4 (Cary, NC, USA).

## 3. Results

Overall, 75 patients who met the inclusion criteria were retrieved from our archives and selected for a retrospective analysis.

According to different protocol time, 31 (41.3%) patients received GnRHa + LNG-IUD, 20 (26.7%) MA + LNG-IUD, and 24 (32%) MA + LNG-IUD + MET, respectively.

Principal characteristics of the study population are detailed by treatment in Table 1.

Mean age at diagnosis was 34.7 ± 5.5 years; women treated with MA + LNG-IUD + MET were significantly older (36.3 ± 5.0 years) than women receiving GnRHa + LNG-IUD (32.8 ± 4.9 years) (*p* = 0.02).

Mean age at menarche and Body Mass Index (BMI) were 12.2 ± 1.6 years and 25.2 ± 6.7, respectively.

Fourteen women (18.7%) were diagnosed with PCOS, most of whom were treated with GnRHa + LNG-IUD during the first study period (*p* = 0.003). Nineteen women (25.3%) were infertile.

Ovarian cancer was found in 13 patients (17.3%), and all of these cases were stage I endometrioid ovarian cancers: 8 out of 13 were synchronous and 5 were metachronous, respectively. Only two women underwent demolitive surgery due to the diagnosis of ovarian carcinoma. The remaining patients were very young and motivated to preserve fertility; consequently, they undertook a conservative management of both ovarian and endometrial neoplasia, after extensive counselling.

Overall, 34 women (44%) were subjected to hysterectomy; the major indications were SD/PD (20/33, 61%) and no more wish to conceive (9/33, 27%). We also reported 4 demolitive surgeries because of metachronous ovarian cancer (2/33, 6%) or other non-oncological causes (2/33, 6%).

In women undergoing hysterectomy for non-oncological indications, no evidence of tumor was found in the uterine specimens. No deaths were reported in the study population.

No comparisons of continuous or categorical patients’ characteristics by treatment protocol were statistically significant, except for age at diagnosis and PCOS. The median follow-up was 45 months (range 6–180).

### 3.1. Oncological Outcomes

After 12 months of FST, CR, PR, SD, and PD occurred in 74.7%, 5.3%, 17.3%, and 2.7% of patients, respectively. Even if not significant (*p* = 0.65), an increasing trend of CR and a diminishing trend of SD were observed according to hormonal therapy (Table 2).

A global increase in relapse CI was found over time and was more frequently detected in patients treated with GnRHa + LNG-IUD, when compared to women treated with MA + LNG-IUD or MA + LNG-IUD + MET, but not significantly (*p* = 0.21) (Figure 1).

Furthermore, there was a not-significant increasing trend of relapse CI over time in women who did not have pregnancy after FST, whereas post-treatment pregnancy did not affect CI (*p* = 0.19) (Figure 2).

At Cox regression analysis, no factors significantly influenced hazard ratio (HR) for time to relapse (Table 3).

Interestingly, age at diagnosis and type of hormonal therapy reached a *p*-value next to significant level; thus, both factors were considered for multivariate analysis.

At univariate analysis, age at diagnosis increased relapse risk (HR = 1.09, 95% CI: 0.99–1.20; *p* = 0.07), whereas MA + LNG-IUD (HR = 0.64, 95% CI: 0.23–1.77; *p* = 0.39) and MA + LNG-IUD + MET (HR = 0.32, 95% CI: 0.10–1.08; *p* = 0.06) decreased HR for time to relapse, when compared to GnRHa + LNG-IUD.

At multivariate analysis, age at diagnosis showed a significantly higher HR (1.13, 95% CI: 1.02–1.25; *p* = 0.02) and MA + LNG-IUD + MET a significantly lower HR (0.22, 96% CI: 0.07–0.68; *p* = 0.009) as opposed to GnRHa + LNG-IUD.

### 3.2. Obstetric Outcomes

Overall, 38 women (50.7%) tried to conceive after reaching CR. Among them, 18 patients became pregnant (24%) with a total of 28 pregnancies and a pregnancy rate of 73.7%. Eighteen out of thirty-three women (54.5%) resorted to medical-assisted conception, including ovarian stimulation.

Outcomes of pregnancies included 9 miscarriages (spontaneous abortion rate of 32.1%) and 19 live births (live birth rate of 42.1%) (Table 1). Ten women had spontaneous vaginal delivery (52.6%), whereas nine underwent Caesarean section (47.4%). The median neonatal weight was of 3100 g (range: 1100–3820).

The median time from the end of treatment to pregnancy was 4 months (range: 1–48).

There was no significant distribution of pregnancy number and outcomes among different treatment groups (Table 1).

Pregnancy complications included one ectopic pregnancy, two twin-pregnancies (one due to in vitro fertilization and one after natural conception), one pre-eclampsia with intrauterine growth restriction, and one preterm birth due to preterm premature rupture of membranes.

At logistic regression analysis, post-treatment pregnancy seemed to reduce relapse risk (HR = 0.45, 95% CI: 0.54–3.32) but this effect did not reach a statistically significant value (*p* = 0.16) (Table 3).

## 4. Discussion

Our study confirms efficacy and safety of FST in women with EC who meet the inclusion criteria for this management, after an adequate pre-treatment selection and regardless of the treatment regimen used. The treatment protocol with MA + LNG-IUD + MET showed a significantly lower HR for time to relapse as opposed to GnRHa + LNG-IUD. However, the statistical power of this finding is limited because this comparison only reached a *p*-value next to significant level at univariate analysis, probably due to the small number of relapse events and the small size of enrolled population. Although our multivariate analysis is not significant, it is interesting to note a lower risk of recurrence in women treated with MA + LNG-IUD + MET versus patients treated with MA + LNG-IUD, when both compared to GnRHa + LNG-IUD regimen. In our analysis, we found a significant difference of PCOS distribution among treatment protocol groups. However, this difference is the result of a randomly greater concentration of patients affected by PCOS during the first period, in which the adopted therapeutic regimen was GnRHa + LNG-IUD.

We found good obstetric outcomes (pregnancy rate 74% and live birth rate 42%) in women who tried to conceive after reaching CR, with no differences among regimens.

Our institutional protocols for FST of EC have changed over years. The first therapeutic approach consisted of GnRHa + LNG-IUD [14]. However, few studies have confirmed the efficacy of this association [15,16,17].

To our knowledge, there are no previous studies comparing the association of LNG-IUD with either GnRHa or oral progestin.

Our study showed comparable efficacy in terms of CR at 12 months, achieved in 74.7% of women. Although not statistically significant, our data show a positive trend with the use of the different therapeutic regimens in terms of CR increase (64.5% with GnRHa + LNG-IUD versus 80.0% with MA+LNG-IUD and 83.3% with MA + LNG-IUD + MET), and a negative trend in terms of SD evidence (25.8% with GnRHa + LNG-IUD versus 15.0% with MA + LNG-IUD and 8.3% with MA + LNG-IUD + MET).

The use of MA + LNG-IUD + MET confirms its better efficacy also at multivariate analysis. Moreover, 50% (10/20) of patients who underwent hysterectomy for PD or SD were treated with GnRHa + LNG-IUD, and women treated with MA + LNG-IUD + MET were significantly older.

We demonstrated that older age at diagnosis significantly increased relapse risk, as previously shown by Chen et al. but not confirmed by the meta-analysis of Koskas et al. Even this result has a limited statistical power for the same reasons discussed above for what concerns different hormonal therapeutic regimens [18,19].

To date, several progestin regimens, way and duration of administrations have been used in FST, but none seems to be preferable [20].

MPA and MA are the most common oral progestins used in FST protocols [21]. Even though there is no consensus in the literature, MA showed a higher remission rate and a lower progression probability than MPA [18]. Side effects of high-dose prolonged oral administration of progestins can be partially bypassed through intrauterine administration. LNG-IUD has been shown to be effective alone in reversing to normal histology in most cases of early-stage EC [22,23]. Moreover, as shown by Kim et al., the combined treatment (LNG-IUD plus oral progestin) is more efficacious in terms of CR when compared to either modality alone [24].

Based on this evidence, another therapeutic regimen, based on the combined use of LNG-IUD + MA, was incorporated into our clinical practice in 2012.

Since many encouraging data have been published on the use of MET in FST of EC and tolerability of its side effects, MET was added to our protocol in 2017. As in our series, MET showed an additive effect to progestins, improving the CR rate and reducing the recurrence rate, especially in overweight women (BMI ≥ 25 kg/m^2^) but also in normal weight patients [10,25].

The mean BMI of our study population was 25.2 ± 6.7, which may have contributed to the better performance of the therapeutic regimen including MET.

Although the desire for pregnancy is an essential requirement for FST, only 50.7% of our patients tried to conceive after reaching CR.

The reasons may be related to the idea that infertility has implications that go beyond the immediate reproductive needs of patients, because gynecological cancers and subsequent radical surgeries have a strong impact on the female identity, sexuality, and body image. Adequate psychological support should be offered to all women wishing FST [26,27].

The fertility outcome analysis of our study population shows a high pregnancy rate, despite a higher incidence of infertility before cancer diagnosis (25.3%), than in the general population (10–15%) [28].

Instead, the spontaneous abortion rate was 42.1%, that is considerably higher than previously reported in the general population (15–20%) [29].

Nevertheless, our finding appears less worrying considering that 18 women resorted to medical-assisted conception and that spontaneous abortion rate rises to 30% by adding the pre-clinical pregnancy loss rate in the general population [30].

Although not statistically significant, the decreased risk of recurrence found in our population with post-treatment pregnancy is in line with other authors’ findings. Indeed, two Korean studies showed a better disease-free survival in women who had at least one pregnancy after FST compared to those who did not [31,32].

Strengths of the present study are the rigorous selection of patients, the long period of follow-up and the comparison of different hormonal therapeutic regimens. However, limitations include its retrospective and single center design and the small sample size.

## 5. Conclusions

FST of EC is a safe and effective option for women who desire to preserve fertility by all hormonal regimens analyzed. No differences were found among the different therapies in terms of obstetric outcomes. The addition of MET to progestin therapy seems to reduce the risk of recurrence and the GnRHa + LNG-IUD combined therapy remains an effective option when a high-dose progestin treatment is not recommended.

## Figures and Tables

**Figure 1 cancers-14-02170-f001:**
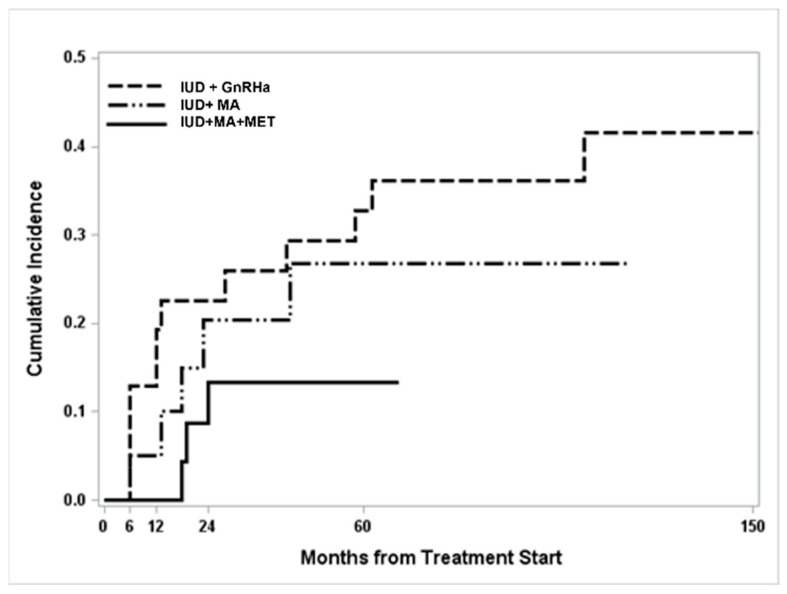
Cumulative incidence curves for relapse according to treatment.

**Figure 2 cancers-14-02170-f002:**
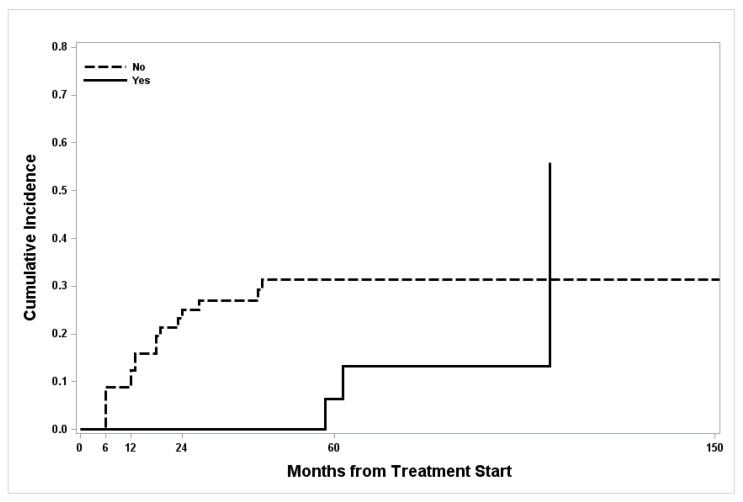
Cumulative incidence curves for relapse according to post-treatment pregnancy status.

**Table 1 cancers-14-02170-t001:** Patient’s characteristics, categorical and continuous variables summary statistics by treatment.

Characteristic			Treatment (LNG-IUD Combinations) N (%) ^a^			
		All Patients N = 75	+GnRHa N = 31 (41.3%)	+MA N = 20 (26.7%)	+MA/MET N = 24 (32.0%)	*p*-Value ^b^
**PCOS**		14 (18.7)	11 (35.5)	0	3 (12.5)	**0.003**
**Hypertension**		3 (4.0)	1 (3.2)	1 (5.0)	1 (4.2)	1.00
**Diabetes**		1 (1.3)	1 (3.2)	0	0	1.00
**Endometriosis**		12 (16.2)	4 (12.9)	5 (25.0)	3 (13.0)	0.48
**Infertility**		19 (25.3)	8 (25.8)	4 (20.0)	7 (29.2)	0.80
**Ovarian cancer**		13 (17.3)	4 (12.9)	6 (30.0)	3 (12.5)	0.25
**Hysterectomy**		34 (45.3)	18 (58.1)	8 (40.0)	8 (33.3)	0.18
**Smoke**	**Never**	50 (74.6)	23 (74.2)	15 (79.0)	12 (70.6)	
	**Former**	11 (16.4)	6 (19.4)	1 (5.3)	4 (23.5)	
	**Present**	6 (9.0)	2 (6.5)	3 (15.8)	1 (5.9)	0.47
**Try to conceive**		38 (50.7)	14 (45.2)	10 (50.0)	14 (58.3)	0.62
**Post-treatment pregnancy**		18 (24.0)	8 (25.8)	5 (25.0)	5 (20.8)	0.94
**Infertility treatment, N = 33**		18/33 (54.5)	5/10 (50.0)	5/10 (50.0)	8/13 (61.5)	0.82
**Delivery number** **N = 18**	**0**	2 (11.1)	1/8 (12.5)	0	1/5 (20.0)	
	**1**	13 (72.2)	4/8 (50.0)	5/5 (100)	4/5 (80.0)	
	**2**	3 (16.7)	3/8 (37.5)	0	0	0.31
**Miscarriages ^c^**		7 (9.3)	3 (9.7)	2(10.0)	2 (8.3)	1.00
			**Treatment (LNG-IUD Combinations)** **Mean (SD)**			
**Characteristic**	**All Patients** **N = 75**		**+GnRHa** **N = 31** **(41.3%)**	**+MA** **N = 20** **(26.7%)**	**+MA/MET** **N = 24** **(32.0%)**	***p*-Value ^d^**
**Age at diagnosis, years**	34.7 (5.5)		32.8 (4.9)	34.5 (6.4)	36.3 (5.0)	**0.04 ^e^**
**Age at menarche, years N = 67**	12.2 (1.6)		12.2 (2.0)	12.3 (1.3)	12.0 (1.3)	0.74
**Body Mass Index N = 72**	25.2 (6.7)		25.8 (8.1)	23.8 (4.5)	25.6 (6.2)	0.82

^a^ Column % except for PMA and Delivery number as indicated; ^b^ Fisher’s exact test; ^c^ Min = 1, Max = 3; polycystic ovary syndrome (PCOS); ^d^ Kruskal–Wallis test or type III fixed effects F-test (overall between treatments comparisons); ^e^ Pairwise comparisons: GnRH analogue vs. megestol/metformin, *p* = 0.02; GnRH analogue vs. megestrol, *p* = 0.55; megestrol vs. megestrol/metformin, *p* = 0.69.

**Table 2 cancers-14-02170-t002:** Treatment response at 12 months from treatment start.

		Treatment (LNG-IUD Combinations) N (Column %)			
**Response**	**All Patients** **N = 75**	**+GnRHa** **N = 31**	**+MA** **N = 20**	**+MA/MET** **N = 24**	***p*-Value**
**CR**	56 (74.7)	20 (64.5)	16 (80.0)	20 (83.3)	
**PR**	4 (5.3)	2 (6.5)	1 (5.0)	1 (4.2)	
**SD**	13 (17.3)	8 (25.8)	3 (15.0)	2 (8.3)	
**PD**	2 (2.7)	1 (3.2)	0	1 (4.2)	0.65

Complete response (CR), partial response (PR), stable disease (SD), progression of disease (PD).

**Table 3 cancers-14-02170-t003:** Cox regression Hazard Ratio (HR) estimates for time (months) to relapse.

Risk Factor	Level	Univariate		Multivariate	
		HR (95% CI)	*p*-Value	HR (95% CI)	*p*-Value
**Age at diagnosis**		1.09 (0.99, 1.20)	0.07	1.13 (1.02, 1.25)	**0.02**
**PCOS**	**No**	Ref		-	
	**Yes**	1.34 (0.54, 3.32)	0.53	-	
**Post-treatment pregnancy**	**No**	Ref		-	
	**Yes**	0.45 (0.54, 3.32)	0.16	-	
**Treatment** **(LNG-IUD combinations)**	**+GnRHa** **+MA**	Ref 0.64 (0.23, 1.77)	- 0.39	Ref 0.50 (0.19, 1.31)	- 0.16
	**+MA/MET**	0.32 (0.10, 1.08)	0.06	0.22 (0.07, 0.68)	**0.009**

## Data Availability

The data presented in this study are available on request from the corresponding author. The data are not publicly available due to patients’ privacy restrictions. The data are safely stored in a private database of the European Institute of Oncology, Milan, Italy.

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
