# Peer review of "Fertility-Sparing Treatment for Endometrial Cancer: Oncological and Obstetric Outcomes in Combined Therapies with Levonorgestrel Intrauterine Device"

_cancers, 2022, doi:10.3390/cancers14092170_

Round 1
Reviewer 1 Report
Authors described that the treatment protocol with MA+LNG-IUD+MET showed a significantly lower HR for time to relapse as opposed to GnRHa+LNG-IUD. However, the statistical power of this finding is limited because this comparison only reached a p- value next to significant level at univariate analysis, probably due to the small number of relapse events and the small size of enrolled population. Therefore, they concluded that no differences were found among the different therapies in terms of obstetric outcomes. The addition of MET to progestin therapy seems to reduce the risk of recurrence and the GnRH analogue + LNG-IUD combined therapy remains an effective option when a high dose progestin treatment is not recommended.
As authors described, this article does not conclude the difference between the therapies mainly because the small number of enrolled population. To make a better massage by the manuscript, authors should try to have further enrollment of patients.
Author Response
REVIEWER 1
Authors described that the treatment protocol with MA+LNG-IUD+MET showed a significantly lower HR for time to relapse as opposed to GnRHa+LNG-IUD. However, the statistical power of this finding is limited because this comparison only reached a p- value next to significant level at univariate analysis, probably due to the small number of relapse events and the small size of enrolled population. Therefore, they concluded that no differences were found among the different therapies in terms of obstetric outcomes. The addition of MET to progestin therapy seems to reduce the risk of recurrence and the GnRH analogue + LNG-IUD combined therapy remains an effective option when a high dose progestin treatment is not recommended.
As authors described, this article does not conclude the difference between the therapies mainly because the small number of enrolled population. To make a better massage by the manuscript, authors should try to have further enrollment of patients.
We thank the reviewer for his comments and suggestions. We are aware that the statistical power of our findings is limited, but we already enrolled patients for a long period (17 years) after a rigorous pre-treatment selection. Moreover, we excluded from final analysis all patients that changed protocol therapy or did not respect the scheduled follow-up controls.
The reviewer is certainly right in pointing out the need for a larger study population, however we can’t increase the sample size in reasonable short time frame because of the rarity of the pathology in young women.
Although not statistically significant, our data show a positive trend with the use of the different therapeutic regimens in terms of CR increase (64.5% with GnRHa+LNG-IUD versus 80.0% with MA+LNG-IUD and 83.3% with MA+LNG-IUD+MET) and a negative trend in terms of SD evidence (25.8% with GnRHa+LNG-IUD versus 15.0% with MA+LNG-IUD and 8.3% with MA+LNG-IUD+MET).
As recommended, we performed a revision of the English language. All corrections are highlighted in yellow in the text.

Reviewer 2 Report
It is a very interesting and exciting retrospective analysis. The manuscript is well written and structured. The research topic discussed about patients with fertility-sparing treatment for endometrial cancer with an inspiring result of fertility-sparing treatment.
Since it's a retrospective analysis, is there any reason or expert opinion to mention the choice of these three different treatments in different time periods?
Is primary diagnosis including hysteroscopy, D and C of endometrium or only office endometrium sampling biopsy?
May you consider to analyze those who failed the treatment and compare between the different subgroups of patients?
Reviewer 3 Report
This is a retrospective study which aims to compare the different fertility-Sparing treatment regimens for endometrial cancer.
Recently, there was a study performed in the Spain which compared the oncological, fertility and obstetric outcomes in endometrial cancer patients who accepted fertility-Sparing treatment with the levonorgestrel intrauterine device (LNG-IUD), megestrol acetate or medroxyprogesterone acetate (DOI: 10.1007/s00404-021-06375-2).
The current study compared 3 different approaches: GnRH analogue + LNG-IUD Vs Megestrol acetate (MA) + LNG-IUD Vs MA + LNG-IUD + Metformin (MET) , which is quite similar to the design of the above published study. Thus the novelty is not that high. Unfortunately, the current study is not sufficient to get published in such a journal of Cancers.
Round 2
Reviewer 1 Report
Authors described that the treatment protocol with MA+LNG-IUD+MET showed a significantly lower HR for time to relapse as opposed to GnRHa+LNG-IUD. However, the statistical power of this finding is limited because this comparison only reached a p- value next to significant level at univariate analysis, probably due to the small number of relapse events and the small size of enrolled population. Therefore, they concluded that no differences were found among the different therapies in terms of obstetric outcomes. The addition of MET to progestin therapy seems to reduce the risk of recurrence and the GnRH analogue + LNG-IUD combined therapy remains an effective option when a high dose progestin treatment is not recommended.
As authors described, this article does not conclude the difference between the therapies mainly because the small number of enrolled population. To make a better massage by the manuscript, authors should try to have further enrollment of patients.
It is hard to increase the sample size in reasonable short time frame, however, the paper has to give a clear and correct massages to readers by this journal. Authors should try the other journals.
Author Response
We thank the reviewer for his comments. We are aware about the limits of our study. However, our paper does not provide confounding endpoints. As highlighted by the Editor, the rigorous pre-treatment selection and the strict adherence to the protocol therapy give value to our long lasting experience in the management of a so rare pathology in young women.

Reviewer 3 Report
Thanks for the respond, however, there are still several limitations listed below needs further explanations.
- As was mentioned in the respond from the authors, the current study compared 3 combined regimens with LNG-IUD: GnRHa + LNG-IUD in 31 pts - low dose MA + LNG-IUD in 20 pts and MA + MET + LNG-IUD in 24 pts, however, there were still certain number of patients who are not suitable for IUDs.
- In the currents study, patients from different period were included (• GnRHa+LNG-IUD, from to February 2005 to September 2012; • MA 160 mg/day+LNG-IUD, from October 2012 to September 2017; • MA 160 mg/day+LNG-IUD+MET 500 mg 3 times per day (1500 mg/day), from October 2017 to October 2021). I am wondering whether different time period would affect the Pregnancy rate and live birth rate?
- In total, 75 patients were included in the current study. There were 13 patients with synchronous or metachronous ovarian cancer. In my opinion, ovarian cancer patients should be excluded in this study.
Author Response
We thank you for your valuable comments and suggestions. Please find below my point-by-point replies to your concerns.
- In our series none of the patients showed absolute contraindications to IUDs (PID, congenital uterine abnormalities or fibroids distorting the cavity in a manner incompatible with proper IUD placement). Furthermore, we performed all hysteroscopic guided endometrial biopsies under sedation, which definitely helps to make the method more widely applicable.
- The reviewer is probably right in pointing out the impact of the time period on the obstetric outcomes, also considering the progress of IVF in the last few years and the length of our recruitment. From a statistical point of view, there is no meaningful difference between group comparison because of the lack of statistical power due to the small number of women with post-treatment pregnancy in each group (8 in GnRHa+LNG-IUD - 5 in MA+LNG-IUD and 5 in MA+MET+LNG-IUD).
- The ovarian cancer diagnosis did not greatly affect the course of the FST for endometrial cancer in our series. All the reported cases were stage I endometrioid ovarian cancers: 8/13 were synchronous and 5/13 were metachronous. Among them, 6/13 women underwent to demolitive surgery: 4 for PD/SD or recurrence of endometrial neoplasia and only 2 for the ovarian cancer. The first case was a synchronous stage Ia ovarian cancer which relapsed after 5 years of follow-up. The second was a metachronous stage Ia ovarian carcinoma in a 42-year-old patient, that was found out two years after the first diagnosis of endometrial cancer. Patients who did not undergo to demolitive surgery (7/13) were highly motivated to perform a FST and chose a conservative management for both the ovarian and the endometrial cancers, after an extensive counseling. Currently, they carry out the follow-up protocol at our Institute with NED and a mean follow up of 74.8 months (range: 6 -154). In addition, cases of synchronous or metachronous ovarian cancer are present in all treatment groups (6 in GnRHa + LNG-IUD - 4 in MA + LNG-IUD and 3 in MA + MET + LNG-IUD). The reviewer rightly noticed this poorly explained aspect of our study, so we have provided more details about patients affected by synchronous or metachronous ovarian cancer in the Results section (lines 165-170).
